# A Probabilistic State Space Model for Joint Inference from Differential Equations and Data

**Jonathan Schmidt**
University of Tübingen
Tübingen, Germany
jonathan.schmidt@uni-tuebingen.de

**Nicholas Krämer**
University of Tübingen
Tübingen, Germany
nicholas.kraemer@uni-tuebingen.de

**Philipp Hennig**
University of Tübingen
Max Planck Institute for Intelligent Systems
Tübingen, Germany
philipp.hennig@uni-tuebingen.de

## Abstract

Mechanistic models with differential equations are a key component of scientific applications of machine learning. Inference in such models is usually computationally demanding, because it involves repeatedly solving the differential equation. The main problem here is that the numerical solver is hard to combine with standard inference techniques. Recent work in probabilistic numerics has developed a new class of solvers for ordinary differential equations (ODEs) that phrase the solution process directly in terms of Bayesian filtering. We here show that this allows such methods to be combined very directly, with conceptual and numerical ease, with latent force models in the ODE itself. It then becomes possible to perform approximate Bayesian inference on the latent force as well as the ODE solution in a single, linear complexity pass of an extended Kalman filter / smoother — that is, at the cost of computing a single ODE solution. We demonstrate the expressiveness and performance of the algorithm by training, among others, a non-parametric SIRD model on data from the COVID-19 outbreak.

## 1 Introduction

Mechanistic models based on ordinary differential equations (ODEs) are popular across a wide range of scientific disciplines. To increase the descriptive power of such models, it is common to consider parametrized versions of ODEs and find a set of parameters such that the dynamics reproduce empirical observations as accurately as possible. Algorithms for this purpose typically involve repeated forward simulations in the context of, e.g., Markov-chain Monte Carlo or optimization. The need for iterated computation of ODE solutions may demand simplifications in the model to meet limits in the computational budget.

This work describes an algorithm that merges mechanistic knowledge in the form of an ODE with a non-parametric model over the parameters controlling the ODE – a *latent force* that represents quantities of interest. The algorithm then infers a trajectory that is informed

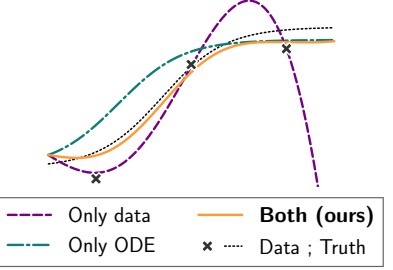

Figure 1: Inferring an unknown function with a Gaussian Process and different sources of information.

35th Conference on Neural Information Processing Systems (NeurIPS 2021).

by the observations but also follows sensible dynamics, as defined by the ODE, in the absence of observations (Figure 1). The main insight enabling this approach is that if probabilistic ODE solvers use the language of (extended) Kalman filters, conditioning on observations and solving the ODE itself is possible in one and the same process of Bayesian filtering and smoothing. Instead of iterated computation of ODE solutions, a posterior distribution arises from *a single* forward simulation, which has complexity equivalent to numerically computing an ODE solution, once, with a filtering-based, probabilistic ODE solver [32]. Intuitively, one can think of this as opening up the black box ODE solver and acknowledging that each task – solving the ODE and discovering a latent force – is probabilistic inference in a state-space model.

The main contribution of this work is formalizing this intuition. Several experiments empirically prove the efficiency and the expressivity of the resulting algorithm. In particular, a practical model for the dynamics of the COVID-19 pandemic is considered, in which a non-parametric latent force captures the effect of policy measures that continuously change the contact rate among the population.

## 2 Problem setting

Let $x : [t_0, t_{\max}] \to \mathbb{R}^d$ be a process that is observed at a discrete set of points $\mathcal{T}_N^{\text{OBS}} := (t_0^{\text{OBS}}, ..., t_N^{\text{OBS}})$ through a sequence of measurements $y_{0:N} := (y_0, ..., y_N) \in \mathbb{R}^{(N+1) \times k}$. Assume that these measurements are subject to additive i.i.d. Gaussian noise, according to the observation model

$$y_n = Hx(t_n) + \epsilon_n, \quad \epsilon_n \sim \mathcal{N}(0, R), \tag{1}$$

for $n = 0, ..., N$ and matrices $H \in \mathbb{R}^{k \times d}$ and $R \in \mathbb{R}^{k \times k}$. Further suppose that $x(t)$ solves the ODE

$$\dot{x}(t) = f(x(t); u(t)), \tag{2}$$

and satisfies the initial condition $x(t_0) = x_0 \in \mathbb{R}^d$. The vector field $f : \mathbb{R}^d \times \mathbb{R}^\ell \to \mathbb{R}^d$ is assumed to be autonomous, which is no loss of generality (e.g. [19]) but simplifies the notation. The *latent force* $u : [t_0, t_{\max}] \to \mathbb{R}^\ell$ parametrizes $f$ and shall be unknown.

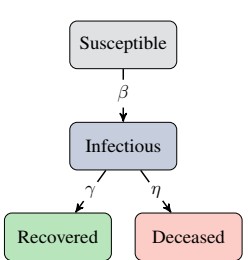

Figure 2: SIRD dynamics.

SIR-type models (e.g. [7]) are a common choice to describe the evolution of the COVID-19 pandemic. In SIR-type models, a population partitions into a discrete set of compartments. The differential equation then describes the transition of counts of individuals between these compartments. For example, the SIRD model [10] formulates the transitions between susceptible, infectious, recovered, and deceased people as

$$
\begin{aligned}
\dot{S}(t) &= -\beta(t)S(t)I(t)/P, & \dot{R}(t) &= \gamma I(t), \\
\dot{I}(t) &= \beta(t)S(t)I(t)/P - \gamma I(t) - \eta I(t), & \dot{D}(t) &= \eta I(t),
\end{aligned} \tag{3}
$$

governed by contact rate $\beta(t) : [t_0, t_{\max}] \to [0, 1]$, recovery rate $\gamma \in [0, 1]$, and mortality rate $\eta \in [0, 1]$ (Figure 2). $S$, $I$, $R$, and $D$ evolve over time, but the total population $P$ (as the sum of the compartments) is assumed to remain constant. In this context, the contact rate $\beta(t)$ is the latent force and varies over time (in the notation from Eq. (2), $\beta$ is $u$). A time-varying contact rate provides a model for the impact of governmental measures on the dynamics of the pandemic. The experiments in Section 5 isolate the impact of the contact rate on the course of the infection counts, by assuming that $\gamma$ and $\eta$ are fixed and known. The method is by no means restricted to inference over a single latent force, as will also be shown in Section 5.1. In this SIRD setting, the goal is to infer an (approximate) joint posterior over $\beta(t)$ and the dynamics of $S(t)$, $I(t)$, $R(t)$, and $D(t)$ as well as to use the reconstructed dynamics to extrapolate into the future. Section 3 explains the conceptual details, Section 4 distinguishes the method from related work, and Section 5 evaluates the performance.

## 3 Method

This section explains how to infer the unknown process $u(t)$ and the ODE solution $x(t)$ in a single forward solve. Section 3.1 defines the prior model, Section 3.2 describes the probabilistic numerical ODE inference setup, and Section 3.3 describes approximate Gaussian filtering and smoothing in this context. Section 3.4 summarizes the resulting algorithm. The exposition of classic concepts here is necessarily compact. In-depth introductions can be found, e.g., in the book by Särkkä and Solin [28].

## 3.1 Gauss–Markov prior

Let $\nu \in \mathbb{N}$. Define two independent Gauss–Markov processes $\mathrm{U} : [t_0, t_{\max}] \to \mathbb{R}^\ell$ and $\mathrm{X} : [t_0, t_{\max}] \to \mathbb{R}^{d(\nu+1)}$ that solve the linear, time-invariant stochastic differential equations [25],

$$d\mathrm{U}(t) = F_\mathrm{U}\mathrm{U}(t)\,dt + L_\mathrm{U}\,d\mathrm{W}_\mathrm{U}(t), \quad d\mathrm{X}(t) = F_\mathrm{X}\mathrm{X}(t)\,dt + L_\mathrm{X}\,d\mathrm{W}_\mathrm{X}(t), \tag{4}$$

with drift matrices $F_\mathrm{U} \in \mathbb{R}^{\ell \times \ell}$ and $F_\mathrm{X} \in \mathbb{R}^{d(\nu+1) \times d(\nu+1)}$, as well as dispersion matrices $L_\mathrm{U} \in \mathbb{R}^{\ell \times s}$ and $L_\mathrm{X} \in \mathbb{R}^{d(\nu+1) \times d}$. $\mathrm{W}_\mathrm{U} : [t_0, t_{\max}] \to \mathbb{R}^s$ and $\mathrm{W}_\mathrm{X} : [t_0, t_{\max}] \to \mathbb{R}^d$ are Wiener processes. $\mathrm{U}$ and $\mathrm{X}$ satisfy the Gaussian initial conditions,

$$\mathrm{U}(t_0) \sim \mathcal{N}(m_\mathrm{U}, P_\mathrm{U}), \quad \mathrm{X}(t_0) \sim \mathcal{N}(m_\mathrm{X}, P_\mathrm{X}), \tag{5}$$

defined by $m_\mathrm{U} \in \mathbb{R}^\ell$, $P_\mathrm{U} \in \mathbb{R}^{\ell \times \ell}$, $m_\mathrm{X} \in \mathbb{R}^{d(\nu+1)}$, and $P_\mathrm{U} \in \mathbb{R}^{d(\nu+1) \times d(\nu+1)}$. $\mathrm{U}(t)$ models the unknown function $u(t)$ and can be any Gauss–Markov process that admits a representation as the solution of a linear SDE with Gaussian initial conditions. $\mathrm{X}(t) = (\mathrm{X}^{(0)}(t), ..., \mathrm{X}^{(\nu)}(t)) \in \mathbb{R}^{d(\nu+1)}$ models the ODE dynamics, in light of which we require $\mathrm{X}^{(i)}(t) = \frac{d^i}{dt^i}\mathrm{X}^{(0)}(t) \in \mathbb{R}^d$, $i = 0, ..., \nu$. In other words, the first element in $\mathrm{X}(t)$ is an estimate for $x(t)$, the second element is an estimate for $\frac{d}{dt}x(t)$, et cetera. Encoding that the state $\mathrm{X}$ consists of a model for $x(t)$ as well as its first $\nu$ derivatives imposes structure on $F_\mathrm{X}$ and $L_\mathrm{X}$ (see e.g. [18]). Examples include the Matérn, integrated Ornstein-Uhlenbeck, and integrated Wiener processes; the canonical choice for probabilistic ODE solvers would be integrated Wiener processes [29, 32, 4, 19].

The class of Gauss–Markov priors inherits its wide generalizability from Gaussian process models; recall that Gauss–Markov processes like $\mathrm{U}$ and $\mathrm{X}$ are Gaussian processes with the Markov property. While not every Gaussian process with one-dimensional input space is Markovian, a large number of descriptions of Gauss–Markov processes emerge by translating a covariance function into an (approximate) SDE representation [28, Chapter 12]. For example, this applies to (quasi-)periodic, squared-exponential, or rational quadratic kernels; in particular, sums and products of Gauss–Markov processes admit a state-space representation [30, 28]. Recent research has considered approximate SDE representations of general Gaussian processes in one dimension [20]. With these tools, prior knowledge over $\mathrm{U}$ or $\mathrm{X}$ can be encoded straightforwardly into the model.

## 3.2 Two likelihoods: for observations and for the ordinary differential equation

A functional relationship between the processes $\mathrm{U}(t)$, $\mathrm{X}(t)$ and the data $y_{0:N}$ emerges by combining two likelihood functions: one for the observations $y_{0:N}$ (recall Equation (1)), and one for the ordinary differential equation. The present section formalizes both. Let $\mathcal{T} = \mathcal{T}_N^{\mathrm{OBS}} \cup \mathcal{T}_M^{\mathrm{ODE}}$ be the union of the observation-grid $\mathcal{T}_N^{\mathrm{OBS}}$, which has been introduced in Section 2, and an ODE-grid $\mathcal{T}_M^{\mathrm{ODE}} := (t_0^{\mathrm{ODE}}, ..., t_M^{\mathrm{ODE}})$. The name "ODE-grid" expresses that this grid contains the locations on which the ODE information will enter the inference problem, as described below.

$\mathcal{T}_N^{\mathrm{OBS}}$ contains the locations of $y_{0:N}$, in light of which the first of two observation models is

$$\mathrm{Y}_n \mid \mathrm{X}(t_n^{\mathrm{OBS}}) \sim \mathcal{N}\left(H\mathrm{X}^{(0)}(t_n^{\mathrm{OBS}}), R\right), \tag{6}$$

for $n = 0, \ldots, N$. This is a reformulation of the relationship between process $x$ and observations $y_{0:N}$ in Eq. (1) in terms of $\mathrm{X}$ (instead of $x$, which is modeled by $\mathrm{X}^{(0)}$). Including this first measurement model ensures that the inferred solution remains close to the data points. $\mathcal{T}_M^{\mathrm{ODE}}$ contains the locations on which $\mathrm{U}(t)$ connects to $\mathrm{X}(t)$ through the ODE. Specifically, the set of random variables $\mathrm{Z}_{0:M} \in \mathbb{R}^{(M+1) \times d}$, defined as

$$\mathrm{Z}_m \mid \mathrm{X}(t_m^{\mathrm{ODE}}), \mathrm{U}(t_m^{\mathrm{ODE}}) \sim \delta\left(\mathrm{X}^{(1)}(t_m^{\mathrm{ODE}}) - f\left(\mathrm{X}^{(0)}(t_m^{\mathrm{ODE}}); \mathrm{U}(t_m^{\mathrm{ODE}})\right)\right), \tag{7}$$

where $\delta$ is the Dirac delta, describes the discrepancy between the current estimate of the derivative of the ODE solution (i.e. $\mathrm{X}^{(1)}$) and its desired value (i.e. $f(\mathrm{X}^{(0)}; \mathrm{U})$), as prescribed by the vector field $f$. If the random variables $\mathrm{Z}_{0:M}$ realize small values everywhere, $\mathrm{X}^{(0)}$ solves the ODE as parametrized by $\mathrm{U}$. This motivates introducing artificial data points $z_{0:M} \in \mathbb{R}^{(M+1) \times d}$ that are equal to zero, $z_m = 0 \in \mathbb{R}^d$, $m = 0, ..., M$. Conditioning the stochastic processes $\mathrm{X}$ and $\mathrm{U}$ on attaining this (artificial) zero data ensures that the inferred solution follows ODE dynamics throughout the domain. Figure 3 shows the discretized state-space model.

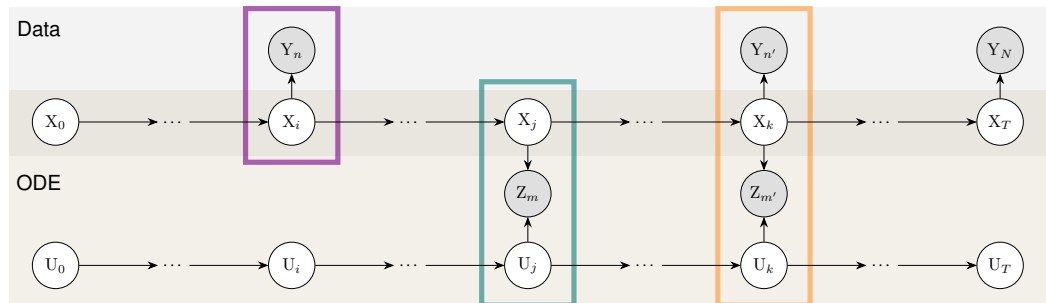

Figure 3: **Instance of the described state-space model, visualized as a directed graphical model**. Shaded variables are observed. Either *only data*, *only mechanistic knowledge*, or *both sources of information* can be conditioned on during inference (recall Figure 1).

### 3.3 Approximate inference with an extended Kalman filter

Both X and U enter the likelihood in Eq. (7) through a possibly non-linear vector field $f$. Therefore, the posterior distribution (recall $z_{0:M} = 0$)

$$p\big(\mathrm{U}(t), \mathrm{X}(t) \mid \mathrm{Z}_{0:M} = z_{0:M}, \, \mathrm{Y}_{0:N} = y_{0:N}\big) \tag{8}$$

is intractable, but can be approximated efficiently. Even though the problem is discretized, the posterior distribution is continuous [28, Chapter 10]. There are mainly two approaches to computing a tractable approximation of the intractable posterior distribution in Eq. (8): approximate Gaussian filtering and smoothing [27], which computes a cheap, Gaussian approximation of this posterior, and sequential Monte Carlo methods [24], whose approximate posterior may be more descriptive, but also more expensive to compute. Like the literature on probabilistic ODE solvers [32, 4], this work uses approximate Gaussian filtering and smoothing techniques for their low computational complexity.

The continuous-discrete state-space model inherits its non-linearity from the ODE vector field $f$. Linearizing $f$ with a first-order Taylor series expansion creates a tractable inference problem; more specifically, it gives rise to the extended Kalman filter (EKF) [13, 22]. Loosely speaking, if the random variable Z is large in magnitude, then X and U are poor estimates for the ODE and its parameter. An EKF update, based on the first-order linearization of $f$, approximately corrects this misalignment. If sufficiently many ODE measurements $z_{0:M}$ are available, a sequence of such updates preserves sensible ODE dynamics over time. An alternative to a Taylor-series linearization is the unscented transform, which yields the unscented Kalman filter [34, 15]. The computational complexity of both algorithms is linear in the number of grid points and cubic in the dimension of the state-space. Detailed implementation schemes can be found, for instance, in the book by Särkkä [27].

The EKF approximates the filtering distribution

$$p\left(\mathrm{U}(t), \mathrm{X}(t) \mid \mathrm{Z}_{0:m} = z_{0:m}, \mathrm{Y}_{0:n} = y_{0:n}, \text{ such that } t_m^{\mathrm{ODE}}, t_n^{\mathrm{OBS}} \le t\right). \tag{9}$$

It describes the current state of the system given all the previous measurements and allows updates in an online fashion as soon as new observations emerge. If desired, the Rauch-Tung-Striebel smoother turns the filtering distribution into an approximation of the full (smoothing) posterior (in Eq. (8)). In doing so, all observations – that is, measurements according to both Eq. (6) and Eq. (7) – are taken into account for the posterior distribution at each location $t$. As special cases, this setup recovers: (i) a Kalman filter/Rauch-Tung-Striebel smoother [16] if the ODE likelihood (Eq. (7)) is omitted; (ii) an ODE solver [32], if the data likelihood (Eq. (6)) is omitted. In the present setting, however, both likelihoods play an important role.

### 3.4 Algorithm and implementation

The procedure is summarized in Algorithm 1. The prediction step is determined by the prior and is available in closed-form (Appendix A.2). At times at which data is observed according to the linear Gaussian measurement model in Eq. (6), the update step follows the rules of the standard Kalman filter. Before updating on pseudo-observations according to the ODE likelihood (Eq. (7)), the non-linear measurement model is linearized at the predicted mean. More details are provided

**Algorithm 1** Compute the filtering distribution by conditioning on both $y_{0:N}$ and $z_{0:M}$.

---

**Input:** data $y_{0:N}$, time grid $\mathcal{T} = \mathcal{T}_N^{\text{OBS}} \cup \mathcal{T}_M^{\text{ODE}}$, vector field $f$, $m_{\text{X}}$, $P_{\text{X}}$, $m_{\text{U}}$, $P_{\text{U}}$
**Output:** Filtering distribution                 [Eq. (9)]
Initialize $\text{X}_0 = \mathcal{N}(m_{\text{X}}, P_{\text{X}})$ and $\text{U}_0 = \mathcal{N}(m_{\text{U}}, P_{\text{U}})$        [Eq. (5)]
**for** $t_j \in \mathcal{T}$ **do**
  Predict $\text{X}_j$ from $\text{X}_{j-1}$ and predict $\text{U}_j$ from $\text{U}_{j-1}$
  **if** $t_j \in \mathcal{T}_N^{\text{OBS}}$ **then** update $X_j$ on $y_j$ **end if**        [Eq. (6)]
  **if** $t_j \in \mathcal{T}_M^{\text{ODE}}$ **then** linearize measurement model and update $X_j$ and $\text{U}_j$ on $z_j$ **end if**  [Eq. (7)]
**end for**

---

in Appendix A. The filtering distribution can be turned into a smoothing posterior by running a backwards-pass with a Rauch-Tung-Striebel smoother (e.g. [27]).

The computational cost of obtaining either, the filtering or the smoothing posterior, are both linear in the number of grid points and cubic in the dimension of the state-space, i.e. $\mathcal{O}((N + M)(d^3\nu^3 + \ell^3))$. Only a single forward-backward pass is required. If desired, the approximate Gaussian posterior can be refined iteratively by means of posterior linearization and iterated Gaussian filtering and smoothing, which yields the maximum-a-posteriori (MAP) estimate [2, 31]. The experiments presented in Section 5 show how a single forward-backward pass already approximates the MAP estimate accurately.

## 4 Related work

**Latent forces and ODE solvers:** The explained method closely relates to probabilistic ODE solvers and latent force models [37], especially the kind of latent force model that exploits the state-space formulation of the prior [9]. The difference is that, in the spirit of probabilistic numerical algorithms, the mechanistic knowledge in the form of an ODE is injected through the likelihood function instead of the prior. A similar approach of linking observations to mechanistic constraints has previously been used in the literature on constrained Gaussian processes [14] and gradient matching [5, 36]. Probabilistic ODE solvers have been used by Kersting et al. [17] for efficient ODE inverse problem algorithms, but their approach is different to the present algorithm, in which the need for iterated optimization or sampling is avoided altogether.

**Monte Carlo methods:** (Markov-chain) Monte Carlo methods are also able to infer a time-dependent ODE latent force from a set of state observations. Options that are compatible with a setup similar to the present work would include sequential Monte Carlo techniques [24], elliptical slice sampling [23], or Hamiltonian Monte Carlo [3] (for instance realized as the No-U-Turn sampler [12]). The shared disadvantage of Monte Carlo methods applied to the resulting ODE inverse problem is that the complexity of obtaining *a single* Monte Carlo sample is of the same order of magnitude as computing the *full* Gaussian approximation of the posterior distribution. In Appendix B we show results from a parametric version of the SIRD-latent force model (using the No-U-Turn sampler as provided by NumPyro [26]). This sampler requires *thousands* of numerical ODE solutions, compared to the single solve of our method. This fact is also reflected in the wall-clock time needed for both types of inference. While the MCMC experiment in Appendix B takes in the order of hours, each experiment with our approach takes under one minute to complete. In other words, the algorithm in the present work poses an efficient yet expressive alternative to Monte Carlo methods for approximate inference with dynamical systems.

## 5 Experiments

This section describes three blocks of experiments. The implementation is based on ProbNum [35] and all experiments use a conventional, consumer-level CPU. First, a range of artificial datasets is generated by sampling ODE parameters from a prior state-space model and simulating a solution of the corresponding ODE. Inference in such a controlled environment allows comparing to the ground truth, thereby assessing the quality of the approximate inference. We consider three ODE models to this end. Second, a COVID-19 dataset will probe the predictive performance of the probabilistic model and the resulting approximate posterior distribution. Third, some changes to the model from the COVID-19 experiments, for instance, ensuring that the number of case counts must be positive,

will improve the interpretability (for example, of the credible intervals). Controlling the range of values that the prior state-space can realize introduces additional non-linearity into the model – which can also be locally approximated by the EKF – and makes the solution more physically meaningful.

## 5.1 Simulated environments

As a first test for the capabilities of the proposed method, we consider three simulated environments. To this end, the training data is generated as follows. The starting point is always an initial value problem with dynamics defined by a vector field $f$ and a Gauss–Markov prior over the dynamics $x$ and the unknown parameters $u$ of the vector field. Then, (i) we sample the time-varying parameter trajectories from the Gauss–Markov prior; (ii) we solve the ODE, as parametrized by the sampled trajectories from (i), using LSODA [11] with adaptive step sizes using SciPy [33]; (iii) we subsample the ground-truth solution on a uniform grid (which will become $\mathcal{T}_N^{\text{OBS}}$) to generate artificial state observations $y_{0:N}$; (iv) we add Gaussian i.i.d. noise to the observations.

The procedure described above generates both a ground truth to compare to and a noisy, artificially observed data set. Given such a set of observations, Algorithm 1 computes a posterior distribution over the true trajectories under appropriate model assumptions. In this posterior, we look for the proximity of the mean estimate to the underlying ground truth; the closer, the better. We measure this proximity in the root-mean-square error. Furthermore, the width of the posterior (expressed by the posterior covariance) should deliver an appropriate quantification of the mismatch. We report the $\chi^2$-statistic [1], which suggests that the posterior distribution is well-calibrated if the $\chi^2$-statistic is close to the dimension $d$ of the ground truth. Three mechanistic models serve as examples.

**Van-der-Pol:** The first of three test problems is the van-der-Pol oscillator [8]. It has one parameter $\mu$ (sometimes referred to as a stiffness constant, because for large $\mu$, the van-der-Pol system is stiff). As a prior over the dynamics we choose a twice-integrated Wiener process with diffusion intensity $\sigma_{\text{X}}^2 = 300$. The stiffness parameter $\mu$ is modeled as a Matérn-³⁄₂ process with lengthscale $\ell_{\text{U}} = 10$ and diffusion intensity $\sigma_{\text{U}}^2 = 0.3$. The posterior is computed on a grid from $t_0 = 0$ to $t_{\max} = 25$ units of time with step size $\Delta t = 0.025$.

**Lotka-Volterra:** The Lotka-Volterra equations [21] describe the change in the size of two populations, predators and prey. There are four parameters, which we call $a$, $b$, $c$, and $d$, which describe the interaction and death/reproduction rates of the populations. As a prior over the dynamics we choose a twice-integrated Wiener process with diffusion intensity $\sigma_{\text{X}}^2 = 10$. The four parameters are modeled as Matérn-³⁄₂ processes with lengthscales $\ell_{\text{U}_a} = \ell_{\text{U}_b} = \ell_{\text{U}_c} = \ell_{\text{U}_d} = 40$. The diffusion intensities are $\sigma_{\text{U}_a}^2 = \sigma_{\text{U}_c}^2 = 0.01$ and $\sigma_{\text{U}_b}^2 = \sigma_{\text{U}_d}^2 = 0.001$. The posterior is computed on a grid from $t_0 = 0$ to $t_{\max} = 60$ units of time with step size $\Delta t = 0.1$.

**SIRD:** As detailed in Section 2, the SIRD model is governed by a contact rate $\beta(t)$. Recall that we assume a time-dependent $\beta$ to account for governmental measures in reaction to the spread of COVID-19. The recovery rate $\gamma$ and fatality rate $\eta$ are fixed at $\gamma = 0.06$ and $\eta = 0.002$, like they will be in the experiments with real data in Sections 5.2 and 5.3 below. As a prior over the dynamics we choose a twice-integrated Wiener process with diffusion intensity $\sigma_{\text{X}}^2 = 50$. The contact rate $\beta$ is modeled as a Matérn-³⁄₂ process with lengthscale $\ell_{\text{U}} = 14$ and diffusion intensity $\sigma_{\text{U}}^2 = 0.1$. The posterior is computed on a grid from $t_0 = 0$ to $t_{\max} = 100$ units of time with step size $\Delta t = 0.1$.

The model allows for straightforward restriction of parameter values by using link functions. The natural support for the SIRD-contact rate is the interval $[0, 1]$, but $U(t)$, as a Gauss–Markov process, takes values on the real line. A change in the basis of $\beta(t)$ with a logistic sigmoid function $\vartheta$ before it enters the likelihood fixes this misspecification. Similarly, the Lotka-Volterra parameters are inferred in log-space to ensure positivity. It is an appealing aspect of the EKF that these non-linear transformations do not require significant adaptation of the algorithm. Instead, the EKF treats it as merely another level of linearization of Eq. (7). Section 5.3 extends this to the state dynamics.

The results are shown in Figure 4. On all test problems, the algorithm recovers the true states and the true latent force accurately. The recovery is not exact, which shows how the Gaussian posterior is only an approximation of the true posterior. The $\chi^2$-statistic for the van-der-Pol stiffness parameter $\mu$ is 1.11, which lies in $(0.0039, 3.8415)$, the 90% confidence interval of the $\chi^2$ distribution with 1 degree of freedom. The root-mean-square error (RMSE) to the truth is 0.14. The $\chi^2$-statistic for the Lotka-Volterra parameters is 8.06, which lies in $(0.7107, 9.4877)$, the 90% confidence interval of the $\chi^2$ distribution with 4 degrees of freedom. The RMSE to the truth is 0.04 in log space and 0.018 in

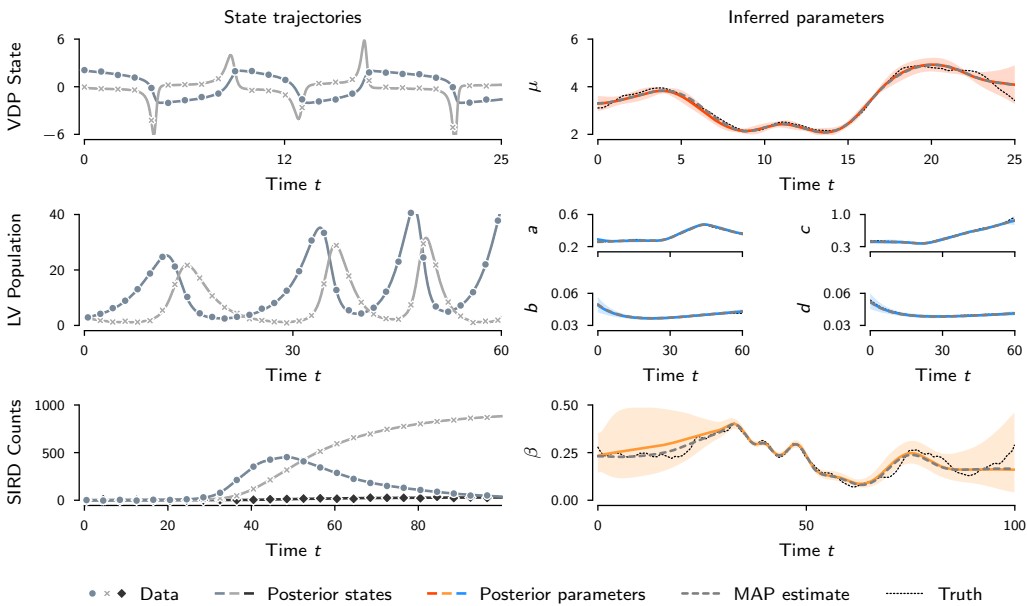

Figure 4: **State recovery in simulated environments**. The stiffness parameter of the van-der-Pol oscillator (top row) and the Lotka-Volterra parameters (middle row) are inferred accurately with appropriately high certainty. For the SIRD experiment (bottom row), the uncertainty is high, where low case counts provide little information about the latent contact rate. With more fluctuations in the observed counts, the approximated contact rate displays more certainty.

linear space. The $\chi^2$-statistic for the contact rate $\beta$ is $0.91$, which lies in $(0.0039, 3.8415)$, the $90\%$ confidence interval of the $\chi^2$ distribution with $1$ degree of freedom. The RMSE to the truth is $0.2$ in logit space and $0.033$ in linear space.

## 5.2 COVID-19 data

We continue with the SIRD model introduced in Eq. (3), now using data collected in Germany over the period from January 22, 2020, to May 27, 2021. Throughout the pandemic, the German government has imposed mitigation measures of varying severity. Together with seasonal effects, summer vacations, etc., they caused a continual change in the contact rate. The next experiments aim to recover said contact rate (and the SIRD counts) from the German dataset.

The Center for Systems Science and Engineering at the Johns Hopkins University publishes daily counts of confirmed ($y_n^{\text{confirmed}}$), recovered ($y_n^{\text{recovered}}$), and deceased ($y_n^{\text{deceased}}$) individuals [6]. One can transform this data to suit the SIRD model

$$I_n := y_n^{\text{confirmed}} - R_n - D_n, \qquad R_n := y_n^{\text{recovered}}, \qquad D_n := y_n^{\text{deceased}}. \qquad (10)$$

The counts $I_n$, $R_n$, and $D_n$ are available for each day, starting with January 22, 2020. Assuming a constant population over time, the numbers of susceptible individuals $S_n$ are always evident from the other quantities, thus left out of the visualizations. We fix the population at $P = 83\,783\,945$, based on public record. We rescale the data to cases per one thousand people (CPT).

As a prior over $\mathrm{X}(t)$, due to its popularity in constructing probabilistic ODE solvers [32], we assume a twice-integrated Wiener process. $\beta(t)$ is modelled as a Matérn-$^3/_2$ process with length scale $\ell_q = 75$ and diffusion intensity $\sigma_q^2 = 0.05$. The state-space model is straightforwardly extendable to sums and products of (more) processes [30, 28]. Inferring parameters that are constant over time, however, is not straightforward due to potentially singular transition models [27, Section 12.3.1].

As described in Section 5.1, the contact rate is inferred in logit space. We shift the logistic sigmoid function such that it fulfills $\vartheta(0) = 0.1$ in which case the stationary mean $\overline{\mathrm{U}} = 0$ translates to a stationary mean $\vartheta(\overline{\mathrm{U}}) = \overline{\beta} = 0.1$ of the Matérn process that models the contact rate. The recovery

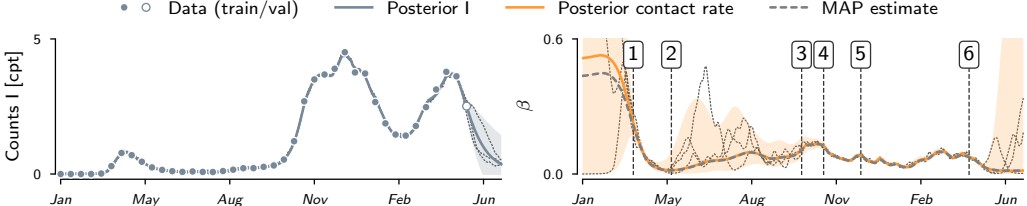

Figure 5: **Estimated counts of infectious cases and contact rate based on real COVID-19 data**. The case counts of infectious people are scaled to cases per thousand (cpt). The uncertainty over the contact rate increases when the case counts are low. After a single forward solve, the inferred mean is already close to the MAP estimate. The shaded areas show the 95 % credible interval and the dotted black lines are samples from the posterior.

Table 1: List of selected governmental measures imposed in Germany with the aim to contain the spread of COVID-19. These events are depicted in Figures 5 and 6 (see column 'Mark'). Links to the sources are provided in Appendix C.

| Mark | Governmental Measures |
|---|---|
| 1 | Stringent contact restrictions, partial shutdown of public life |
| 2 - 3 | Continual relaxations of measures |
| 4 | Partial shutdown of public life ('lockdown light') |
| 5 | Hard lockdown, stringent contact restrictions |
| 6 | First nationwide decree of restrictions, increased intensification of measures |

rate and mortality rate are considered known and fixed at $\gamma = 0.06$ and $\eta = 0.002$ to isolate the effect of the inference procedure on recovering the evolution of the contact rate $U(t) = \beta(t)$. We set the mean of the Gaussian initial conditions to the first data point that is available. The diffusion intensity of the prior process $X(t)$ is set to $\sigma_X^2 = 10$. The latent process $U$ and all derivatives are initialized at zero. Note that due to the logistic sigmoid transform, an initial value $U_0 = 0$ amounts to an initial contact rate $\beta_0 = 0.1$.

In the present scenario, we cannot take the SIRD model as an accurate description of the underlying data but merely as a tool that aids the inference engine in recovering physically meaningful states and forces. In order to account for this model mismatch, the Dirac likelihood from Eq. (7) is relaxed towards a Gaussian likelihood with measurement noise $\lambda^2 = 0.01$. This equals the data observation noise and thus balances the respective impact of either (misspecified) source of information. Intuitively, adding ODE measurement noise reduces how strictly the vector field dynamics are enforced during inference and therefore avoids overconfident estimates of $\beta(t)$.

The mesh-size of the ODE is $\Delta t = 1/24$ days, i.e. ODE updates are computed on an hourly basis. The final 14 observations are excluded from the training set to serve as validation data for evaluating the extrapolation behavior of the proposed method. Figure 5 shows the results. The mean of the state $X$ estimates the case counts accurately in both interpolation and extrapolation tasks. The estimated contact rate rapidly decreases around late March, remains low until fall, increases momentarily, and is dampened again soon after. This aligns with a set of political measures imposed by the government (compare Figure 5 to Table 1). The uncertainty over the estimated contact rate is high in the early beginning when the case counts are still low. It then increases again in summer and with the beginning of the extrapolation phase.

If the experiment is taken as-is, the credibility intervals of the posterior over $X(t)$ include negative numbers (mostly where the case counts are low and the uncertainty high, and when extrapolating). Of course, in a system that models counts of people in different stages of a disease, negative numbers should be excluded altogether. The proposed method provides straightforward means to address this issue. Section 5.3 explains the details.

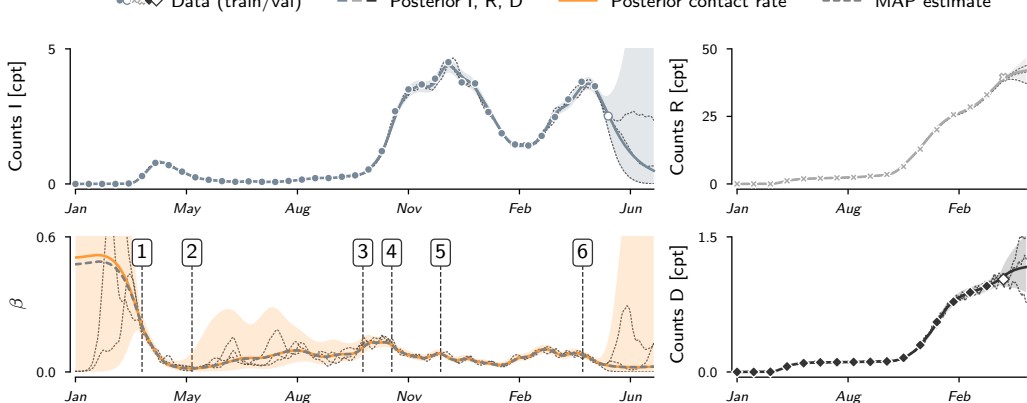

Figure 6: **Estimated case counts and contact rate**, inferred in the logarithmic basis on real COVID-19 and vaccination data. The case counts of infectious people are scaled to cases per thousand (cpt). Again, the uncertainty of the contact rate increases where the case counts are low. Now, the posterior credible interval is restricted to the positive reals. The shaded areas show the 95 % credible interval and the dotted black lines are samples from the posterior.

### 5.3 Non-negative state estimates

The following experiment evaluates how the proposed method performs in combination with a state-space model that constrains the support of the dynamics. Concretely, let $X(t)$ model the logarithm of the SIRD dynamics and the respective derivatives. With a slight abuse of notation, we will continue writing "X" even though it lives in a different space than in the previous sections. The structure of the dynamic model is the same. The diffusion intensity of the prior process $X(t)$ is $\sigma_X^2 = 0.05$. The diffusion is not comparable to the value in the previous section because the state dynamics moved to log-space. Using $\frac{\mathrm{d}}{\mathrm{d}t}\exp(x(t)) = \exp(x(t))\dot{x}(t)$, the ODE likelihood becomes

$$Z_m \mid X_m^{\text{ODE}}, U_m^{\text{ODE}}, \sim \mathcal{N}(\zeta_1 - f(\zeta_2; \zeta_3), \lambda^2 I_d), \tag{11}$$

with auxiliary quantities (recall the logistic sigmoid $\vartheta$)

$$\zeta_1 := \exp\left(X^{(0)}(t_m^{\text{ODE}})\right) X^{(1)}(t_m^{\text{ODE}}), \quad \zeta_2 := \exp\left(X^{(0)}(t_m^{\text{ODE}})\right), \quad \zeta_3 := \vartheta(U(t_m^{\text{ODE}})). \tag{12}$$

The exponential function introduces an additional non-linearity into the state-space model, which necessitates smaller step-sizes for the ODE measurements (see below).

The observed case count data $y_{0:N}$ is transformed into the log-space, too, in which we assume additive, i.i.d. Gaussian noise. On the one hand, transforming the measurements into log-space implies that the measurement model for the counts remains linear; on the other hand, it imposes a log-normal noise model (if viewed back in "linear space"). Log-normal noise underlines how the estimated states cannot be negative. Again, we scale the counts to cases per thousand.

As depicted in Figure 6, the reconstruction of the driving processes in this setting yields results that at first glance, look similar to the previous experiment. The states match the data points well. However, the extrapolation is more realistic in that the credible intervals encode that negative values are impossible (which is due to the log-transform). The mean of the recovered contact rate closely resembles the estimate of the previous experiment. Again, upon implementation of strict governmental measures, the uncertainty decreases, whereas in the context of relaxations, the uncertainty is high.

## 6 Statement on Societal Impact

This work performs methods research to develop an efficient numerical algorithm to infer latent forces governing ordinary differential equations. As a testbed, we use data from the COVID-19 pandemic. We do so to motivate and visualize the practical value of our methods. The results of this

algorithm, however, should not be taken as policy advice. The model used in the paper is deliberately simplistic. The presented work therefore should not be misunderstood as epidemiological research. The machine learning community has, over time, frequently used data of contemporary societal concern to motivate and test new algorithmic concepts (well-known examples from the UCI collection include the Wisconsin Breast Cancer Dataset, the mushroom classification dataset, and the German credit data set). Our work follows in this line. Of course, if this algorithm, or any competitor, is used to derive policy advice, the underlying differential equation and latent states must be carefully considered by domain experts, which we are not.

# 7    Conclusion

By coupling mechanistic and data-driven inference so directly, the algorithm builds on the core premise of probabilistic numerics – that computation itself is a data source that does not differ, formally, from observational data. Information from observations and mechanistic knowledge (in the form of an ODE) can thus be described in the same language of Bayesian filtering and smoothing. This removes the need for an outer loop over multiple forward solves and thus drastically reduces the computational cost. Our experimental evaluation corroborates that the resulting approximate posterior is close to the ground truth and drastically reduces computational cost over Monte Carlo alternatives. It faithfully captures multiple sources of uncertainty from the data, numerical (discretization) error, and epistemic uncertainty about the mechanism. We hope this framework helps empower practitioners, not just by reducing computational burden but also by providing a more flexible modelling platform.

## Acknowledgements

The authors gratefully acknowledge financial support by the European Research Council through ERC StG Action 757275 / PANAMA; the DFG Cluster of Excellence "Machine Learning - New Perspectives for Science", EXC 2064/1, project number 390727645; the German Federal Ministry of Education and Research (BMBF) through the Tübingen AI Center (FKZ: 01IS18039A); and funds from the Ministry of Science, Research and Arts of the State of Baden-Württemberg. The authors thank the International Max Planck Research School for Intelligent Systems (IMPRS-IS) for supporting N. Krämer. Moreover, the authors thank Nathanael Bosch and Marvin Pförtner for valuable discussions.

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
