# Appendix: A Probabilistic State Space Model for Joint Inference from Differential Equations and Data

**Jonathan Schmidt**
University of Tübingen
Tübingen, Germany
jonathan.schmidt@uni-tuebingen.de

**Nicholas Krämer**
University of Tübingen
Tübingen, Germany
nicholas.kraemer@uni-tuebingen.de

**Philipp Hennig**
University of Tübingen
Max Planck Institute for Intelligent Systems
Tübingen, Germany
philipp.hennig@uni-tuebingen.de

## A    Implementation details

This section provides detailed information about the state-space model and approximate Gaussian inference therein. Appendix A.1 defines the augmented state-space model that formalizes the dynamics of the Gauss–Markov processes introduced in Section 3.1. Appendix A.2 provides the equations for prediction and update steps of the extended Kalman filter in such a setup, which is described in Section 3.4 (in particular, Algorithm 1).

### A.1    Augmented state-space model

Section 3 describes the joint inference of both a latent process $u(t) : [t_0, t_{\max}] \to \mathbb{R}^l$ that parametrizes an ODE and $x(t) : [t_0, t_{\max}] \to \mathbb{R}^d$, the solution of said ODE. The dynamics of the processes are modeled by the stochastic differential equation

$$\mathrm{d} \begin{pmatrix} \mathrm{U}(t) \\ \mathrm{X}(t) \end{pmatrix} = \underbrace{\begin{pmatrix} F_{\mathrm{U}} & 0 \\ 0 & F_{\mathrm{X}} \end{pmatrix}}_{=:F} \begin{pmatrix} \mathrm{U}(t) \\ \mathrm{X}(t) \end{pmatrix} \mathrm{d}t + \underbrace{\begin{pmatrix} L_{\mathrm{U}} & 0 \\ 0 & L_{\mathrm{X}} \end{pmatrix}}_{=:L} \mathrm{d} \begin{pmatrix} \mathrm{W}_{\mathrm{U}}(t) \\ \mathrm{W}_{\mathrm{X}}(t) \end{pmatrix}, \tag{A.1}$$

with Gaussian initial conditions

$$\begin{pmatrix} \mathrm{U}(t_0) \\ \mathrm{X}(t_0) \end{pmatrix} \sim \mathcal{N} \left( \begin{pmatrix} m_{\mathrm{U}}(t_0) \\ m_{\mathrm{X}}(t_0) \end{pmatrix}, \begin{pmatrix} P_{\mathrm{U}}(t_0) & 0 \\ 0 & P_{\mathrm{X}}(t_0) \end{pmatrix} \right). \tag{A.2}$$

The block-diagonal structure is due to the independent dynamics of the prior processes. The *drift matrices* $F_{\mathrm{U}}$ and $F_{\mathrm{X}}$, as well as the *dispersion matrices* $L_{\mathrm{U}}$ and $L_{\mathrm{X}}$ depend on the choice of the respective processes U and X. The measurement models are given in Eq. (6) (for observed data) and in Eq. (7) (for ODE measurements).

In the experiments presented in Sections 5.2 and 5.3 we model the latent contact rate $\beta(t)$ as a Matérn-$3/2$ process with characteristic length scale $\ell_q$. Hence,

$$\mathrm{d}\mathrm{U}(t) = \underbrace{\begin{pmatrix} 0 & 1 \\ -\left(\sqrt{3}/\ell_q\right)^2 & -2\sqrt{3}/\ell_q \end{pmatrix}}_{F_{\mathrm{U}}} \mathrm{U}(t)\,\mathrm{d}t + \underbrace{\begin{pmatrix} 0 \\ 1 \end{pmatrix}}_{L_{\mathrm{U}}} \mathrm{d}\mathrm{W}_{\mathrm{U}}(t). \tag{A.3}$$

The SIRD counts are modeled as the twice-integrated Wiener process

$$\mathrm{dX}(t) = \underbrace{\begin{pmatrix} 0 & I_d & 0 \\ 0 & 0 & I_d \\ 0 & 0 & 0 \end{pmatrix}}_{F_\mathrm{X}} \mathrm{X}(t)\,\mathrm{dt} + \underbrace{\begin{pmatrix} 0 \\ 0 \\ I_d \end{pmatrix}}_{L_\mathrm{X}} \mathrm{dW_X}(t), \tag{A.4}$$

such that $\mathrm{X} = \left(\mathrm{X}^{(0)}, \mathrm{X}^{(1)}, \mathrm{X}^{(2)}\right)^\top$ models the SIRD counts and the first two derivatives. Notice that $F_\mathrm{X} \in \mathbb{R}^{d(\nu+1)\times d(\nu+1)}$ and $L_\mathrm{X} \in \mathbb{R}^{d(\nu+1)\times d}$ are block matrices. $I_d$ denotes the $d \times d$ identity matrix. In the context of the experiments, $d = 4$ (S, I, R, and D) and $\nu = 2$ (*twice*-integrated Wiener process). More details on the use of integrated Wiener processes in probabilistic ODE solvers can be found in, for instance, the work by Kersting et al. [5].

## A.2 Kalman filter equations

This section is concerned with the exact steps that make up the algorithm summarized in Section 3.4. The stochastic differential equation defined in Eq. (A.1) formalizes the dynamics of the processes $\mathrm{U}(t)$ and $\mathrm{X}(t)$ that model $u(t)$ and $x(t)$, respectively. Define $\Delta t := t_j - t_{j-1} > 0$ for all $t_j = t_1, ..., t_{\max}$. The *transition densities* of U and X are [3]

$$\mathrm{U}(t + \Delta t) \mid \mathrm{U}(t) \sim \mathcal{N}(\Phi_\mathrm{U}(\Delta t)\mathrm{U}(t), Q_\mathrm{U}(\Delta t)), \tag{A.5a}$$

$$\mathrm{X}(t + \Delta t) \mid \mathrm{X}(t) \sim \mathcal{N}(\Phi_\mathrm{X}(\Delta t)\mathrm{X}(t), Q_\mathrm{X}(\Delta t)), \tag{A.5b}$$

where transition matrices $\Phi_\mathrm{U}(\Delta t) \in \mathbb{R}^{\ell\times\ell}$ and $\Phi_\mathrm{X}(\Delta t) \in \mathbb{R}^{d(\nu+1)\times d(\nu+1)}$, as well as the process noise covariances $Q_\mathrm{U}(\Delta t) \in \mathbb{R}^{\ell\times\ell}$ and $Q_\mathrm{X}(\Delta t) \in \mathbb{R}^{d(\nu+1)\times d(\nu+1)}$ are available in closed form and can be computed, for instance, with matrix fraction decomposition [12, 1].

Define the transition matrix and process noise covariance of the process in Eq. (A.1) as

$$\Phi(\Delta t) := \begin{pmatrix} \Phi_\mathrm{U}(\Delta t) & 0 \\ 0 & \Phi_\mathrm{U}(\Delta t) \end{pmatrix}, \qquad Q(\Delta t) := \begin{pmatrix} Q_\mathrm{U}(\Delta t) & 0 \\ 0 & Q_\mathrm{U}(\Delta t) \end{pmatrix}. \tag{A.6}$$

Further, let

$$\begin{pmatrix} \mathrm{U}(t_j) \\ \mathrm{X}(t_j) \end{pmatrix} \sim \mathcal{N}\left(m_j, P_j\right), \tag{A.7}$$

for time points $t_j \in \mathcal{T} = \mathcal{T}^{\mathrm{OBS}} \cup \mathcal{T}^{\mathrm{ODE}}$. The predicted mean and covariance $m_j^-$ and $P_j^-$ are

$$m_j^- = \Phi(\Delta t)\,m_{j-1}, \tag{A.8}$$

$$P_j^- = \Phi(\Delta t)P_{j-1}\Phi(\Delta t)^\top + Q(\Delta t), \tag{A.9}$$

for given initial conditions $m_0$, $P_0$. The prediction step is the same, for both $t_j \in \mathcal{T}^{\mathrm{OBS}}$ and $t_j \in \mathcal{T}^{\mathrm{ODE}}$.

As detailed in Section 3, two different update steps are defined for two kinds of observations. When observing data $y_{0:N}$, i.e. $t_n \in \mathcal{T}^{\mathrm{OBS}}$, the update step follows the rules of a standard Kalman filter. The updated mean $m_n$ and covariance $P_n$ at time $t_n$ are computed as

$$v_n = y_n - Hm_n^-, \tag{A.10}$$

$$S_n = HP_n^- H^\top + R, \tag{A.11}$$

$$K_n = P_n^- H^\top S_n^{-1}, \tag{A.12}$$

$$m_n = m_n^- + K_n v_n, \tag{A.13}$$

$$P_n = P_n^- - K_n S_n K_n^\top. \tag{A.14}$$

The matrices $H$ and $R$ are defined as in Eq. (6) in the paper.

Recall the ODE measurement model from Eq. (7), which we here denote as $h$, as

$$h\left(\begin{pmatrix} \mathrm{U}(t) \\ \mathrm{X}(t) \end{pmatrix}\right) = \mathrm{X}^{(1)} - f\left(\mathrm{X}^{(0)}; \mathrm{U}(t)\right). \tag{A.15}$$

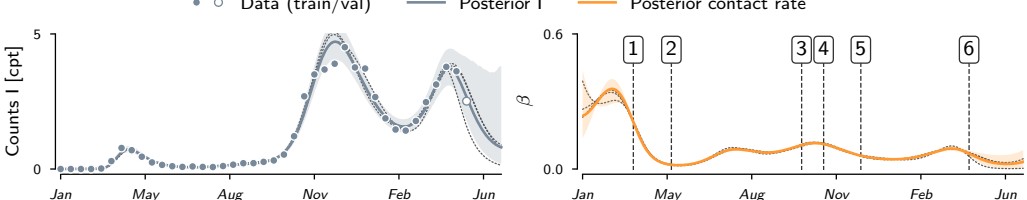

Figure 1: **Estimated counts of infectious cases and contact rate**. The estimates are obtained from MCMC sampling in an SIRD model with a parametric function for the contact rate $\beta(t)$. The case counts of infectious people are scaled to cases per thousand (cpt). The shaded areas show the 95 % credible interval and the dotted black lines are samples from the posterior. Compared to the non-parametric approach presented in the paper, the estimate over $\beta(t)$ is very confident in general. The posterior mean closely resembles the results obtained in Sections 5.2 and 5.3. The numbered markers in the right plot are explained in Table 1 in the paper.

At locations $t_m \in \mathcal{T}^{\text{ODE}}$, we condition on the ODE measurements $z_{0:M}$. Recall that these pseudo-observations are all zero. According to Eq. (10.79) in the book by Särkkä and Solin [11],

$$v_m = z_m - h(m_m^-), \tag{A.16}$$

$$S_m = \left[ \mathrm{D}h(m_m^-) \right] P_m^- \left[ \mathrm{D}h(m_m^-) \right]^\top + \lambda^2 I_d, \tag{A.17}$$

$$K_m = P_m^- \left[ \mathrm{D}h(m_m^-) \right]^\top S_m^{-1}, \tag{A.18}$$

$$m_m = m_m^- + K_m v_m, \tag{A.19}$$

$$P_m = P_m^- - K_m S_m K_m^\top, \tag{A.20}$$

where $[\mathrm{D}h(m_m^-)]$ denotes the Jacobian of $h$ at $m_m^-$. In the case of a Dirac likelihood (see Eq. (7)), $\lambda^2 = 0$ holds. For numerical stability (especially for $\lambda^2 = 0$) one can instead implement square-root filtering (see, e.g., [3, 6]). All experiments in Section 5 use square-root filtering.

## B    Parametric model for MCMC sampling

This section first introduces a functional form for $\beta(t)$ that connects to the non-parametric model introduced in Section 3. Then, a generative model for Markov-chain Monte Carlo (MCMC) inference over the unknown parameters of $\beta(t)$ is set up.

We establish a parametric model for the latent, time-varying contact rate in an SIRD model in terms of Fourier features. In light of Mercer's theorem and the fact that stationary covariance functions have complex-exponential eigenfunctions [10, Chapter 4.3], this closely connects to the Matérn-³⁄₂ process used in Sections 5.2 and 5.3 (see also [9]).

Concretely, we proceed as follows. Let $\mathbb{T}$ denote a dense time grid. First, (i) compute the kernel Gram matrix $K$ on $\mathbb{T}$, such that $(K)_{ij} = k(x_i, x_j)$ with $x_i, x_j \in \mathbb{T}$. $k$ is the Matérn-³⁄₂ covariance function. As in the experiments before, we set the characteristic lengthscale to $\ell = 75$. Then, (ii) compute the eigendecomposition of $K$. In order to keep the dimensionality of the inference problem feasible, select $r \ll |\mathbb{T}|$ eigenvectors that correspond to the $r$ largest eigenvalues of $K$. In this experiment, we choose $r = 25$. (iii) For each eigenvector, the strongest frequency component $\omega$ is determined by the discrete Fourier decomposition. This yields a set of frequencies $\{\omega_i : i = 1, \ldots, r\}$. Finally, the parametric model is defined as the sum of parametrized Fourier features of the form

$$\beta(t) = \vartheta \left( \sum_{i=1}^{r} a_i \cos\left(2\pi\omega_i t\right) + b_i \sin\left(2\pi\omega_i t\right) \right), \tag{B.1}$$

where $\vartheta$ is the logistic sigmoid function as described in Section 5. We aim to compute a posterior contact rate $\beta(t)$ by MCMC inference over the coefficients $a_i$ and $b_i$, $i = 1, \ldots, r$. To this end, we define a prior over the parameter vector $\theta := (a_1, b_1, \ldots, a_r, b_r)^\top$ and a likelihood for the COVID-19 case counts $y_{0:N}$ with respect to $\theta$.

In order to ensure non-negative case counts, as in Section 5.3, we assume log-normally distributed measurements with i.i.d. noise

$$p(y_{0:N} \mid \theta) = \prod_{n=0}^{N} \text{LogNormal}\left(y_n; \log\left(x^{(\theta)}(t_n)\right), \sigma^2 I_{2r}\right),$$  (B.2)

where $\sigma^2$ is inferred from the data along with $\theta$. $x^{(\theta)}(t_n)$ denotes the solution of the SIRD system at time $t_n$, parametrized by the vector of coefficients $\theta$ through the contact rate from Eq. (B.1). Notably, each evaluation of the likelihood involves numerically integrating the SIRD system, which significantly increases the computational cost entailed by the inference algorithm. This is done by NumPyro's DOPRI-5 implementation [8, 2].

The prior distributions over the Fourier-feature coefficients and over $\sigma^2$ are chosen as

$$p(\theta) = \mathcal{N}\left(\theta; \mu_\theta, \Sigma_\theta\right), \qquad p(\sigma^2) = \text{HalfCauchy}(\sigma^2; 0.01).$$  (B.3)

The mean $\mu_\theta$ of the prior over $\theta$ is set to a maximum-likelihood estimate by minimizing the negative logarithm of Eq. (B.2) with SciPy's L-BFGS optimization algorithm [13, 7]. The covariance is chosen as $\Sigma_\theta = 0.1 \cdot I_{2r}$.

The goal of the experiment is to compute a posterior over the coefficients $\theta$ (and the measurement covariance $\sigma^2$) that is comparable to the results obtained in Sections 5.2 and 5.3. Like before, recovery rate and fatality rate are assumed fixed and known at $\gamma = 0.06$ and $\eta = 0.002$. We compute the posterior $p(\theta \mid y_{0:N})$ using NumPyro's implementation of the No-U-Turn sampler [4].

Figure 1 shows the estimated number of infectious people and the contact rate over time as inferred by the MCMC algorithm. The state estimate matches the data points well and the uncertainty increases when extrapolating. Like in the experiments in Sections 5.2 and 5.3, the final 14 observations serve as a validation set and the model extrapolates 31 days into the future. The posterior mean closely resembles the results obtained from our method. However, the uncertainty is lower in general, especially in the beginning and over the summer months.

## C  Sources for governmental measures in Germany

This section provides the sources used to list the governmental measures in Table 1. In order to provide reliable sources, we refer to the official press releases, as published by the German government. For each policy change, we provide a very brief idea of the imposed measures and official sources by the German government (only available in German language).

### C.1  March 22, 2020 (Mark 1)

Citizens are urged to restrict social contacts as much as possible and the formation of groups is sanctioned in public spaces as well as at home.

https://www.bundesregierung.de/breg-de/themen/coronavirus/besprechung-der-bundeskanzlerin-mit-den-regierungschefinnen-und-regierungschefs-der-laender-vom-22-03-2020-1733248

https://www.bundesregierung.de/resource/blob/975226/1733246/e6d6ae0e89a7ffea1ebf6f32cf472736/2020-03-22-mpk-data.pdf?download=1

### C.2  May 6, 2020 (Mark 2)

The government puts the federal states in charge of appropriately relaxing the imposed measures. Different states handle the situation differently, according to the respective incidences (*'hotspot strategy'*).

https://www.bundesregierung.de/breg-de/aktuelles/pressekonferenzen/pressekonferenz-von-bundeskanzlerin-merkel-ministerpraesident-soeder-und-dem-ersten-buergermeister-tschentscher-im-anschluss-an-das-gespraech-mit-den-regierungschefinnen-und-regierungschefs-der-laender-1751050

### C.3  October 7, 2020 (Mark 3) and October 14, 2020

The population is again urged to restrict contacts if possible.

One week later, new light restrictions are imposed. The number of people allowed in social gatherings is limited, according to local incidences.

### C.4 November 2, 2020 (Mark 4)

Partial shutdown of public life (*'lockdown light'*). Across the country, the number of people allowed in social gatherings is limited to ten, where the number of households present must not exceed two. Most of public services are closed or offered only virtually, if possible.

### C.5 December 16, 2020 (Mark 5)

Across the country, the number of people allowed in social gatherings is limited to five, where the number of households present must not exceed two. Except for stores of systemic importance, the retail sector is mostly shut down.

### C.6 April 23, 2021 (Mark 6)

The aforementioned measures were mostly governed and implemented by the respective federal states. On April 22, 2021, the German government decides on a nationwide decree of measures to come into effect on the following day (April 23, 2021). Depending on the seven-day incidence, curfews, contact restrictions, and a shutdown of large parts of public life are imposed.