# OpenReview forum: "A Probabilistic State Space Model for Joint Inference from Differential Equations and Data"
_NeurIPS.cc/2021/Conference — NeurIPS 2021 Poster_

### Official Review · Reviewer_SfYp · 2021-07-08

**Rating:** 6
**Confidence:** 4

**Summary:**

This paper presents a technique for computing approximate ordinary differential equation (ODE) solution posteriors. As shown in previous work [31], the state solutions of mechanistic ODE systems can be approximated by an extended Kalman filter. The authors further extend this methodology by conditioning the filter on observed sequences. Unlike its Monte Carlo alternatives, the resulting inference scheme computes approximate posteriors in a single forward-backward pass. The synthetic data experiments demonstrate that both the state and parameter trajectories are accurately inferred. The vanilla method and a variant that allows non-negative state estimation are tested on a COVID-19 dataset, implying interpretable findings that agree with government measures.

**Limitations And Societal Impact:**

The limitations and potential negative societal impact are addressed. Maybe an additional sentence reminding that the findings are highly related to the assumptions made before inference would be nice.

**Main Review:**

Overall, the paper is written clearly and very easy to follow. The background is explained in sufficient detail. Figures are highly illustrative and communicate the ideas well (figure-1 might be slightly misleading as the curves seem like point estimates while the inference is probabilistic). Both the simulated and COVID experiments nicely show how high level of accuracy and calibration of uncertainty estimates.

To my understanding, the model largely builds upon [31]. Since [31] has already defined the prior process and introduced artificial data points $z_{0:M}$, the main theoretical contributions of the manuscript are (i) the additional prior on $u(t)$ and likelihood, and (ii) the inference in log space. In my opinion, these extensions are straightforward and thus NeurIPS might not be a suitable venue. That being said, I believe the COVID experiments pertain to the challenges we globally face today, which boosts the impact of this manuscript. Therefore, if the NeurIPS organizing committee particularly welcomes COVID-related submissions (about which I couldn't find any information) or I miss any major aspect of the paper, I recommend a reject and re-submission to a more suitable venue. Below are more detailed comments:

- The observation model in eq. (1) is linear, which might be too restrictive for more complicated systems. A discussion on how to extend the method into non-linear observation mappings would be nice.
- The performance of the proposed framework should be analyzed thoroughly. Although the comparison against the MCMC baseline is impressive, I wonder how difficult the addressed problems are. For instance, how does the estimation error change with (intermediate) gaps in the observed sequences? Since the inference is already based on discrete filter update equations, it would be interesting to see how simple discrete-time approximations (to the underlying true ODE system) would behave (this is not to show the merits of continuous-time modeling but to show when/if discrete approximations fail).
- How are the prior parameters selected? An ablation study showing how, e.g., length-scale and M impact the overall performance would be nice. Similarly, does $\lambda$ provide a trade-off between data likelihood and accuracy of ODE solutions?

Minor comments:
- Assuming fixed $\gamma$ and $\eta$ parameters may be too simplistic. In the context of the global pandemic, for instance, different COVID variants and the ratio of people who have developed immunity directly affect these parameters. This should at least be addressed verbally.
- In Figures 5-6, what do the thin gray dashed curves represent (look like the "truth" curve in Figure 4)?
- (minor) The state notation $X(t) = (X^{(0)}(t), ..., X^{(\nu)}(t))$ might be updated to reflect that $X(t)$ is concatenated, e.g., $\mathbf{X}(t) = (X^{(0)}(t), ..., X^{(\nu)}(t))$.

**Post-rebuttal update:** Thanks the authors for the rebuttal. After reading the rebuttal, other reviews/responses, and a little bit of extra thinking, I started to think the paper should be accepted. Utilizing Kalman filters to infer unknown latent forces is indeed one idea that could impact further research and developments. That being said, I have three additional comments that should provide the ground for my final rating:
- The approach assumes a dense observed trajectory. The gradient matching-based "ODE likelihood" (7) would start failing with more and more sparse data. I believe the approach should be stress tested.
- The methodology is tested only on very low-dimensional systems with very simple data likelihood models. So, I wonder how generalizable this framework is.
- The inference heavily depends on [31]. This is not a problem but makes me question the originality of the contribution.

**Time Spent Reviewing:**

6

---

> ### Author Response · Authors · 2021-08-09
> **Author response to Reviewer SfYp**
>
> Dear reviewer,
>
> thank you for taking the time to provide a detailed review. We are glad that you found our paper well-illustrated and clear and that you appreciate the quality of the results of our experiments.
> We hope that we can additionally resolve your main point of criticism in the following and thereby convince you to raise your score.
>
> You are correct that our approach builds upon probabilistic ODE solvers as, for instance, presented in [31].
> However, we argue that this extension is not straightforward at all. The essence of this work lies in providing a shift in perspective, away from black-box ODE solvers and towards a modular state-space formulation that allows for incorporation of multiple (similarly treated) sources of information.
> This is a fundamental change; people tend to think of ODE solvers as the textbook example of black boxes. Our work shows that probabilistic ODE solvers, by explicitly casting the solution of ODEs as state-space inference, can perform inference on latent forces (i.e. a form of "inverse problems") in a single forward solve. Note how doing so blurs the line between "forward" and "inverse" problems, which are widely seen as fundamentally different things.
> In light of this, we are glad that you agree with us on the "high level of accuracy and calibration" of our results.
>
> In the following we want to address specific questions you raised:
> * The linear measurement model in Eq. (1) is by no means a limiting factor. Note that we already use a non-linear measurement model for the ODE observations. It is an appealing aspect of the extended Kalman filter that non-linearities in the state-space model do not result in a considerable increase in the complexity of the method.
> * Due to the assumed underlying dynamics model, interpolation between data points is well informed and guided by the ODE. This improves alternative approaches, like, e.g., standard GP regression. We focused our experiments on extrapolation performance because the available data is very dense over the considered time period.
> * Your intuition aligns with our understanding of $\lambda$. The parameter is indeed used to balance the respective impact of either source of information, i.e. the data and the ODE.
> * Estimating $\gamma$ and $\eta$ as additional latent forces is possible. We consciously decided to isolate the impact of the contact rate on the course of the infection counts. We will add a clarifying remark to the main paper.
> * The thin gray dashed curves you mentioned represent samples from the posterior process. Thank you for raising this concern, we will add this to the respective figure captions.
>
> We hope that these comments have clarified in particular what we see as the substantial contribution of this paper, and will thus convince you to raise your score. If you have any remaining concerns, please do respond to this comment, and we would be glad to discuss things further.
>
> [31] F. Tronarp, H. Kersting, S. Särkkä, and P. Hennig. Probabilistic solutions to ordinary differential equations as nonlinear Bayesian filtering: a new perspective. Statistics and Computing, 29(6): 1297–1315, 2019.

---

### Official Review · Reviewer_RRdP · 2021-07-13

**Rating:** 8
**Confidence:** 2

**Summary:**

The paper "A Probabilistic State Space Model for Joint Inference from Differential Equations and Data" describes a method of how to estimate latent functions in ordinary differential equations using measuremt data on observed parts of the model. This is achieved by combining a (single!) run through an ODE solver with a suitable probabilistic framework (Bayes filter).

**Ethical Concerns:**

Regarding

"Did you discuss any potential negative societal impacts of your work? [No] While the work includes an experimental setting of societal interest (the COVID pandemic), it only serves as an extended example for the utility of the algorithm. It is not used to derive policy recommendations."

One fundamental application of this paper seems to be the usage in modeling spread of disease. This is cause for potential ethical concerns, as political decisions and lives of many people might depend on such models. In particular, I disagree with the authors that in heated online debates such an extended example can not influece policy. However, problems in such a model could lead to real world problems: e.g. detecting changes in viral infectivity too slowly can lull politicians into safety and refusing to take timely countermeasures or e.g. predicting an unreasonaly high rate of infectivity could lead to civil liberties being taken away unnecessary. I have no direct ethical concerns with these models. I even find it helpfull that variance bands are part of the methodology. However, the impact of this paper on society is much higher than for other ML papers, and this should be clearly acknowledged.

I should add that I personally feel very comfortable assessing the technical content of the paper (Gaussian processes, differential equations, solvers, ...), but have no prior knowledge of disease modelling. This justifies my downgrade in reviewer confidence: I cannot bear the responsibility of high confidentiality for such a paper, when lives are at stake and I had no prior point of contact to its application domain.

**Limitations And Societal Impact:**

See ethical concerns.

**Main Review:**

The research question is highly relevant, not only due to COVID-19, the methods are suitable for this problem, and the results are good. The paper is written very clearly, both from the language aspect and from a formal point of view. Suitable experiments are conducted and clearly described. Previous work is cited. This paper is ready to be published at NeurIPS.

Some minor criticism:
 - The title of the paper could be more what kind of differential equations are being used and what is inferred.
 - It might have been helpfull to mention further possible areas of application.
 - Is it possible to work with a non-autonomous differential equation? (lines 49-50)
 - For the wider machine learning audience: What is a latent force? What are latent force models?
 - Due to potential ethical concerns: is the "SIR-type Modell" the canonical/only choice to model COVID-19? What are the alternatives?
 - Line 64: how to estimate gamma and eta might be worth a footnote.
 - How is the ODE grid chosen? equi-distant? adaptive?
 - I would have liked the related work section earlier in the paper.
 - Don't you think that "
    (a) Did you state the full set of assumptions of all theoretical results? [Yes]
    (b) Did you include complete proofs of all theoretical results? [N/A]
   " is a little bit contradictory? I mean one could have said more about function spaces which allow for solutions of the ODEs, existence of solution of ODEs and more aspects of the usual problems when dealing with differential equations. I do not think that this would have helped the paper.

It is a minefield to adequately cite all papers relating differential equations and machine learning. However, the following paper (and references therein) might be suitable to also be cited:
 - Chen et al., Neural ordinary differential equations
 - Dupont et al., Augmented neural odes
 - Jidling et al., Linearly constrained Gaussian processes
 - Lange-Hegermann, Algorithmic linearly constrained Gaussian processes
 - Rubanova et al., Latent odes for irregularly-sampled time series
 - Yang et al., Inference of dynamic systems from noisy and sparse data via manifold-constrained Gaussian processes

**Time Spent Reviewing:**

12

---

> ### Author Response · Authors · 2021-08-09
> **Author response to Reviewer RRdP**
>
> Dear reviewer,
>
> thank you for your strong review. We greatly appreciate that you find our work "highly relevant", "written very clearly", and "ready to be published at NeurIPS".
>
> We first want to address your minor comments before we move on to your ethical concerns.
>
> * Yes, it is possible to work with non-autonomous differential equations. This is detailed, e.g. in [18].
> * SIR-type models are perhaps not the only, but a common and popular choice for describing infectious diseases with differential-equation-based models [10]. We will milden the statement in line 52 accordingly.
> * We do agree with you and reviewer "SfYp" that adding a clarifying remark regarding inference over $\gamma$ and $\eta$ is important, and we will do so.
> * The ODE grid is chosen fixed. This is mostly due to the fact that a dense set of observations is available, which lessens the need for adaptive steps.
> * Indeed, the answers to the checklist appear contradictory. We did specify all simplifying assumptions made as part of the model and the inference algorithm. However, we did not provide theorems to prove.
> * Thank you for providing additional sources that might be related to our work. We will look into how to integrate each of them into the paper, especially the work on linearly constrained Gaussian processes.
>
> We now want to address the ethical concerns you raised:
>
> Thank you for this thoughtful remark. Indeed, the presentation of our method avails itself of a testbed that is emotionally loaded, and understandably so.
> We agree that there is the possibility of a reader misunderstanding and/or misusing our results, something that we consider merely a testbed for our algorithm.
> We did not carelessly choose this terrain as our main example. We do want to highlight the necessity of two major aspects of ML algorithms applied to real-world scenarios, which our method can provide:
> 1. Being honest and clear about uncertainty in predictions where information is sparse. We are glad about your acknowledgment of this aspect in our approach.
> 2. Working towards agile and informed decision making by circumventing expensive iterative computations if possible.
>
> Furthermore, we note that it is not unusual for NeurIPS papers to use issues of contemporary societal concern to motivate and explain the utility of novel algorithmic contributions (e.g. climate modeling with the North American Precipitation dataset, Face Recognition with the Labeled Faces in the Wild, and classics like the Wisconsin Breast Cancer and German Credit Rating datasets, which are even part of the UCI collection and frequently used for purposes completely unrelated to their original motivation).
>
> We fully agree that the fact that this is purely a contribution to machine learning research, and that we do by no means derive or suggest policy recommendations, should have a more prominent place in the main text of the paper. We will make this more explicit and hope that this mitigates the issue that you and other reviewers brought up.
>
> All in all, we greatly appreciate your positive, strong review and we are glad that you enjoyed reading our paper.
>
> [18] N. Krämer and P. Hennig. Stable implementation of probabilistic ODE solvers. arXiv:2012.10106, 2020.
>
> [10] H. W. Hethcote. The mathematics of infectious diseases. SIAM Review, 42(4):599–653, 2000.

---

### Official Review · Reviewer_qgjY · 2021-07-15

**Rating:** 6
**Confidence:** 4

**Summary:**

The paper discusses how to combine data with known differential equations of an observed system, to infer the parameters of these differential equations. To this end, the authors combine the state-space framework with ODE solvers and provide an approximate inference procedure based on extended Kalman filters.

**Limitations And Societal Impact:**

+ It would be nice to provide some details on the Gauss-Markov priors in the appendix.
+ I miss a discussion, how parameters could be estimated in this framework. For example, if some parameters on the system are unknown (but constant), how would you estimate them. I guess, one would think about some EM procedure, like done in standard state-space models. Also, it would help to say what emission parameters H and R (from Eq. 1) are in the experiments. I couldn’t find any details on them in the text, neither, what are observations.
+ I noticed, that the values for the diffusion parameter $\sigma_X^2$ were chosen quite high. How were these parameters selected? Is there a heuristic that can be followed?


**Main Review:**

Though the paper doesn’t provide new techniques itself, the combination of recent developments in probabilistic ODE solvers, and combining them with a state-space framework is interesting for inference of dynamical systems. The paper is mostly well written. Integrating knowledge of the underlying dynamic system into state spaced models, and combining it with observations, is important in many applications. The paper lines out nicely the methodology, that achieves this.
The main critique I have, is that the paper is written very high level, and from the main text it was quite difficult to follow what e.g. is happening in Algorithm 1. I think a bit more detail on the Filter equations couldn’t harm here in Sec. 3.4. Also in the experiments some details, e.g. explicitly writing what is $X(t)$, $U(t)$ and $Y$ for the different cases would help readability in my opinion. Then another important point is that there is no baseline in the experiments. I guess it would be simple to at least show the improvement over a normal state space model, which just uses observations. (Much like it is done schematically in Fig. 1, but providing numbers). Otherwise, I think it is a good paper, and with some changes, it could be even better.


**Time Spent Reviewing:**

4h

---

> ### Author Response · Authors · 2021-08-09
> **Author response to Reviewer qgjY**
>
> Dear reviewer,
>
> Thank you for your positive review! We hope the following addresses your concerns so that you will continue to argue in favor of acceptance, and maybe even more than before.
>
> Let us reply to your main point of critique, which we understood as the paper being "too high level" due to, e.g., a lack of concrete equations backing the textual descriptions.
>
> We understand your concern in that the concrete equations might help the understanding of the described concepts in more detail. However, we decided to focus on an intuitive description and decided to provide the concrete predict- and update steps that constitute the filter in Appendix A.2 (see in particular Eqs. (A.10) through (A.20)).
> We hope that this way the reader can focus on the heart of the conceptual design of the algorithm, instead of the method being hidden behind a stack of equations.
>
> Concerning the baseline in the experiments:
> We actually have results for "normal" state-space model inference available (i.e. your idea of "using just observations"). They are not very interesting, which supports the main point of the paper: Using just the observation operator and not the underlying ODE breaks the link between the observations and the vector field, such that parameter inference becomes impossible. The comparison could thus only be made in the output process (e.g. SIRD-counts) and would neglect the latent parameter process by cutting off the connection to the information extracted from the data.
> In the concrete setting of our experiment, this means the $I$ curve follows the (densely observed) data points closely, but the posterior on the latent force is completely unaffected. It is only when the ODE is considered (i.e. included in the observation model via the information operator) that information propagates to the latent force.
>
> Regarding limitations:
>
> * For details on the Gauss-Markov priors we refer to Appendix A.1.
> * This is indeed an interesting point. Inferring constant parameters in this setting bears challenges on its own (e.g. due to singular transition matrices for constant GM processes). We consider this future work and we will make this more clear by mentioning this issue in the main paper.
> * $\sigma^2$ relates to the output scale of the prior process and its derivatives. If either the state values or their rates of change are high, one might want to choose a larger value for $\sigma$.
>
> All in all, we thank you for your positive assessment. If you like, we would be happy to engage in further discussion below.

---

> > ### Comment · Reviewer_qgjY · 2021-08-19
> > **Thanks for the response**
> >
> > I thank the reviewers for the response. I just wanted to respond to the point of the baseline. I understand, that the main point of the presented methodology is to estimate the time varying parameters of the governing dynamical system. However, in the end, one is often interested in making predictions (also of the output process). So my suggestion/question is whether one could not put a number to this predictive performance, and compare it to some more standard models? So e.g. predict the cases of infections from a certain time point without data. The difference in predictive likelihood between the proposed method and e.g. normal state space models, would give some quantitative motivation to use the proposed methodology (in addition to getting an interpretable dynamical system).
> >
> > Overall, I will stay with my score, because I think the paper would be a nice contribution to the field of state-space modelling.

---

### Official Review · Reviewer_K7YV · 2021-08-04

**Rating:** 7
**Confidence:** 3

**Summary:**

Paper proposes a methodology to combine ODE with latent force models to efficiently solve ODEs and ground them with observed data.

**Limitations And Societal Impact:**

Authors have not even included them. I would first like them to even attempt it before giving any suggestion. They say we do have a model but we don't use it for policy making. This is weird the whole point about this section is authors thinking a bit more constructively about their work rather than passing the buck that it is not our concern. I would encourage authors to at least acknowledge the limitations and assumptions and then say output derived from their work might only be valid under certain specific conditions and any work deriving policy from these estimates should keep them in mind.

**Main Review:**

Authors utilize the recent developments in probabilistic numerics to solve  ODE with a latent force in single forward pass. Authors show the usefulness of their methodology on a non-parametric SIRD model applied to COVID19 spread in Germany.

In general experiments on simulated and real data are done properly and show usefulness of the approach.
The ability to solve ODE model in a single pass is admittedly very useful and something that can make the technique adoptable. However, I would like to see comparisons being done on more sophisticated models and being compared against a MCMC approach in much more detail. Right now authors do a comparison as something of an after thought. Also, all novelty is useful only if first the runtimes for the proposed approach is really short than say NUTS. Secondly, the estimates from the approach are comparable to NUTS or in case of simulated data near to real data.

Finally, it seems everything authors have proposed here is already done in previous works they have referenced apart from specifically applying it to SIRD models. It might be case of misunderstanding on my side as most of the technical details have been taken away from paper and gives me impression that they have been done before. If not, I would at least like to see in SI a full blown description that can justify or outline the exact methodological contribution by authors.

Also, I would like authors to claim less novelty about modeling time-varying contact rate. It is something standard in literature. In case of COVID  I would encourage authors to see work done by Walker, Patrick G. T., et al. "The impact of COVID-19 and strategies for mitigation and suppression in low- and middle-income countries." Science, vol. 369, no. 6502, 24 July 2020, pp. 413-22, doi:10.1126/science.abc0035. The supplement has the model description Which was first published as a pre-print in March 2020. The work along this line is used as a standard model for COVID for quite long. This is essentially a problem with our field that we don't look at work done outside the echo chamber. Even the framework byWalker, Patrick G. T., et al. is opensourced here https://github.com/mrc-ide/squire. Even a more sophisticated version than the author's and Walker, Patrick G. T., et al. is used here https://www.medrxiv.org/content/10.1101/2021.01.11.21249564v1. Which is now a Science Translational Medicine article here DOI: 10.1126/scitranslmed.abg4262. All this models have a time varying contact rate in them.

I have updated my score based on replies from authors and reading comments from other reviewers. My concerns have been mostly meet. So I am updating the score.

**Time Spent Reviewing:**

4

---

> ### Author Response · Authors · 2021-08-09
> **Author response to Reviewer K7YV**
>
> Dear reviewer,
>
> Thank you for your detailed review. We hope the following points sufficiently address your concerns and convince you to raise your score.
>
> ### Increased efficiency of our method over MCMC
>
> We agree with you that the increased efficiency of our method as compared to, for instance, MCMC is most tangible if a significant decrease in wall-clock time can be immediately and consistently observed. And we can assure you, it is.
> For instance, the time needed to run the experiment that led to what is shown in Figure 6 was roughly 40 seconds, whereas the MCMC experiment took around 2 hours to complete (both on a consumer-level CPU).
> We refrained from providing an exact runtime analysis, since exact wall time is implementation-dependent and often (perhaps rightly) criticized as PR.
> However, we agree with you that, in general, the significant runtime improvement is a key advantage of our method and will be made part of the manuscript.
>
> ### Time-dependent contact rate
>
> We are surprised by your statement regarding "claiming novelty about modeling [a] time-varying contact rate". We do at no point present this aspect as part of our contribution - the novelty of our work lies in the efficient inference algorithm for such setups! Choosing the SIRD model as one example, we merely argue why we chose to model $\beta$ as a non-constant function of time, fully aware of the fact that this is not a new idea.
> In fact, since it is perfectly valid to assume a constant contact rate in an SIR-type model, we consider it good practice to reason why we decided against it.
> If your comment relates to a specific paragraph in our paper, we would be grateful if you could point this out to us, so we can formulate this more clearly.
>
>
> ### Societal impact
>
> We take your point that the paper should probably contain an ethics statement, if only to avoid misunderstandings. We will thus add the following paragraph to the main paper:
>
> _This work performs methods research to develop an efficient numerical algorithm to infer latent forces governing ordinary differential equations. As a testbed, we use data from the COVID-19 pandemic. We do so to motivate and visualize the practical value of our methods. The results of this algorithm, however, should not be taken as policy advice. The model used in the paper is deliberately simplistic. The presented work therefore should not be misunderstood as epidemiological research. The machine learning community has, over time, frequently used data of contemporary societal concern to motivate and test new algorithmic concepts (well-known examples from the UCI collection include the Wisconsin Breast Cancer Dataset, the mushroom classification dataset, and the German credit data set). Our work follows in this line. Of course, if this algorithm, or any competitor, is used to derive policy advice, the underlying differential equation and latent states must be carefully considered by domain experts, which we are not._
>
> We are completely in agreement with you that the societal impact of machine learning is considerable, and requires awareness from all sides. At the same time, we hope you will agree that if our community wants to achieve a positive impact on society, we have to develop methods that work on datasets of real-world interest, not just on academic, "sanitized" examples.
>
> We again thank you for taking the time to thoroughly review our paper. If you like, we would be happy to continue the discussion below.

---

> > ### Comment · Reviewer_K7YV · 2021-08-26
> > **Score updated**
> >
> > I would like to first thank you for providing a detailed response to my reviews and to other reviewers. I am more satisfied now than earlier and hence have updated my scores. I will be happy if you can address the following comments which I think are a matter of re-framing:-
> > * Time-dependent contact rate: the issue is that you have not directly stated that \beta(t) is your novelty but way paper is written it goes from the latent force being useful to \beta(t) is better than \beta and this is the only example you give. Hence, it seems you are claiming that. I am happy to concede I was a bit more skeptical in the initial review and agree you might not be claiming it but writing makes it look like that.
> > * Societal Impact: Please be sure to add the above statement in the paper.
> > * Increased efficiency: I agree talking about speed is a bit of PR but stating run-times in a table against say well documented/used tools like Stan/numpyro/pyro for MCMC would be useful.

---

> > > ### Author Response · Authors · 2021-08-27
> > > **Thank you for the update**
> > >
> > > Dear reviewer,
> > >
> > > we are very glad that you found our responses helpful and we greatly appreciate that you raised your score!
> > > We will include the statement regarding societal impact and further address your other suggestions in a revised version of the paper.

---

### Decision · Program_Chairs · 2021-09-27

**Decision:**

Accept (Poster)

**Comment:**

The authors present a method for inferring latent functions in ODE model from observations. Reviewers widely praised the manuscript for clarity and motivation.

The discussion focussed on a technical aspect: that the method only requires a single pass of an ODE solver. This is achieved through a reworking of the problem of ODE solving as a state-space inference problem. this is an exciting insight that could change how the community thinks about solving these sorts of inverse problems.

One reviewer raised a serious concern about the lack of a societal impact statement: the authors have provided their statement in the rebuttal with which I am satisfiedL this must make it into the camera-ready edition.